# Senescent Cells in Early Vascular Ageing and Bone Disease of Chronic Kidney Disease—A Novel Target for Treatment

**DOI:** 10.3390/toxins11020082

**Published:** 2019-02-01

**Authors:** Sam Hobson, Samsul Arefin, Karolina Kublickiene, Paul G. Shiels, Peter Stenvinkel

**Affiliations:** 1Division of Renal Medicine, Department of Clinical Science, Technology and Intervention, Karolinska University Hospital, 14186 Stockholm, Sweden; sam.hobson@ki.se (S.H.); samsul.arefin@ki.se (S.A.); karolina.kublickiene@ki.se (K.K.); 2Institute of Cancer Sciences, MVLS, University of Glasgow, Glasgow G61 1QH, UK; paul.shiels@glasgow.ac.uk

**Keywords:** chronic kidney disease, uremic toxins, senescence, Nrf2, ageing

## Abstract

Together with bone-mineral disorders, premature vascular ageing is a common feature of the uremic phenotype. A detailed understanding of mechanisms involved remains unclear and warrants further research. Available treatment options for end stage renal disease are principally dialysis and organ transplantation, as other treatment alternatives have proven insufficient. Chronic kidney disease (CKD) has been proposed as a model of early vascular and bone ageing, with accumulating evidence supporting the contribution of cellular senescence and the senescence-associated secretory phenotype (SASP) to cardiovascular pathology in CKD. Correspondingly, novel therapies based around the use of senolytic compounds and nuclear factor-erythroid-2-related factor 2 (Nrf2) agonists, have been suggested as attractive novel treatment options. In this review, we detail the contribution of the uremic environment to these processes underpinning ageing and how these relate to vascular health.

## 1. Introduction

Ageing is characterised by a decline in functional physiological capability over the life course. The rate of ageing can be manipulated experimentally, however significant, therapeutic intervention to mitigate physiological decline remains largely limited to pre-clinical models [1]. Ageing is a leading risk factor for multiple chronic ‘burden of life style’ diseases, including cardiovascular disease (CVD), osteoporosis, type-2 diabetes, cancer and chronic kidney disease (CKD), which are responsible for considerable mortality [2]. Baker and Sprott [3] originally postulated that: “*A Biomarker of Aging (BoA) is a biological parameter of an organism that either alone or in some multivariate composite will, in the absence of disease, better predict functional capability at some late age than will chronological age*”. This is relevant to cellular senescence, a state of cellular growth arrest, as the accumulation of senescent cells in a tissue or organ restricts health span (Figure 1) [1]. Cellular senescence is one of the hallmarks of ageing that are common across taxa [1]. It remains unclear, however, whether these hallmarks interact cumulatively, synergistically or independently. Furthermore, these features may underpin age-related morbidities, but may be distinct from the morbidity *per se*. Indeed, it remains undetermined if they act uniformly in context across the life course, or if they are subject to the forces of antagonistic pleiotropy [4,5]. Cellular senescence is also present during mammalian embryonic development, shown to mediate tissue remodelling and cell balance in the mesonephros and endolymphatic sac, respectively [6]. In similarity with oncogene-induced senescence, senescence-associated secretory phenotype (SASP) mediators and cell arrest proteins, p21 and p15, are also expressed. It has been suggested that oncogene-induced senescence is an adaptive mechanism that evolved from developmental senescence [7]. Additionally, in multiple kidney diseases, senescence biomarkers have been observed in renal tissue [8] and more recently these observations have been extended to media calcification within the uremic vasculature [9]. Although further research to better understand potential pathological mechanisms is warranted, it remains to be determined if these reflect intrinsic features of the specific morbidity, or are a consequence of it.

Chronic kidney disease–mineral bone disorder (CKD–MBD) is characterised as a syndrome of mineral abnormalities (e.g., calcium, phosphate, vitamin D, etc.); bone abnormalities (relating to strength, turnover or volume of bone); and vascular calcification (VC), a complication which manifests as early as CKD stage 2 [10]. Coronary artery calcification is an independent predictor of cardiovascular mortality in CKD [11], with over 50% of deaths in CKD patients relating to CVD [12]. This reflects a plethora of detrimental early vascular ageing (EVA) effects, more typical of the vascular environment observed in the elderly [13]. CKD patients present with a progeric phenotype inclusive of increased allostatic load, elevated levels of pro-ageing factors, reduced anti-ageing factors and altered anabolic/catabolic pathways [14,15]. Recent experimental evidence in pre-clinical models has shown that senolytic drugs (which promote effective removal of senescent cells) prolongs the health span and lifespan in mice by up to 36% [16], supporting the initiation of preclinical studies with senolytic drugs in the uremic milieu. Whether this translates into humans remains to be determined [17]. 

In this review, we will appraise current knowledge of VC, focusing on the tunica media and intima arterial calcification, as a complication of CKD–MBD; the consequence of dysregulated mineral metabolism in the uremic milieu, with special reference to elevated serum phosphate levels [18]. We will discuss evidence surrounding the role of cellular senescence in mediating hydroxyapatite crystal deposition and the latest advances in therapeutically targeting VC.

## 2. Senescence

### 2.1. Introduction to Senescence 

In 1961 Hayflick and Moorhead reported human fibroblasts to have a replication doubling potential of 50 (±10) in vitro, otherwise known as the Hayflick limit. They attributed the finding to “senescence at the cellular level” [20], and four years later attributed the replication limit to cellular ageing [21]. The meaning, relevance and implications of this now seminal finding have evolved over the past 50 years. The following section aims to summarise what constitutes cellular senescence, before outlining components specific to CKD–MBD. 

Senescence is characterised as a state of irreversible cellular growth arrest, which acts as an anti-oncogenic mechanism [22]. Senescent cells are metabolically active but not physiologically contributory [8]. Numerous cyto-stressors act as inducers of senescence, including oxidative stress, telomere attrition and mutations [19]. Senescent cells can be grouped into distinct classes: developmental or oncogene-induced; with key differences relating to function and clearance. Developmental senescent cells are induced acutely, and have fundamental roles in tissue growth and patterning [7]. However, their contribution to these processes must be viewed through the lens of antagonistic pleiotropy [5]. Such cells are transiently present; effectively removed by apoptosis [7]. Conversely, oncogene-induced senescent cells are associated with age-related pathologies, driven by their detrimental systemic paracrine effects and inability to be effectively removed [8,23]. These characteristics have been attributed to two important features of senescence; the SASP and the pro-survival senescence cell anti-apoptotic pathways (SCAPs).

### 2.2. SASP

The SASP is composed of pro-inflammatory, pro-fibrotic and matrix degrading components, consisting of cytokines, chemokines, proteases and growth factors [24]. Differences in molecules secreted as part of the SASP have been observed between developmental and oncogene-induced senescent cells, between different cell types and between cells induced by differential stressors [17]. However, interleukin (IL)-1β, IL-6, plasminogen activator inhibitor 1 and monocyte chemoattractant protein 1 are highly conserved SASP components [8]. Age-related morbidities have common underpinning molecular mechanisms [4,5,14], such as persistent low-grade sterile inflammation (“inflammaging”), increased oxidative stress, immunosenescence, mitochondrial dysfunction and metabolic imbalances [25]. How much of the inflammatory component of inflammaging is due to accumulation of senescent cells remains to be proven. Any such relationship may be related to decreased expression of nuclear factor-erythroid-2-related factor 2 (Nrf2), which is a feature of chronic ‘burden of lifestyle’ diseases [26]. In keeping with such a postulate, genetic depletion of Nrf2 has been reported to exacerbate age-related induction of senescence markers and pro-inflammatory SASP factors, resulting in a heightened inflammatory state [27]. However, senescence-associated microRNAs have been shown to mediate downregulation of Nrf2 in endothelial cells [28], implying the relationship between senescence and Nrf2 may be bidirectional.

Another important property of the SASP is its ability to induce senescence in surrounding cells via a bystander effect. This was first demonstrated in vitro, through co-culturing senescent cells with younger cells, and observing the older cells inflicting DNA damage on neighbouring cells [29]. The effect occurs via nuclear factor κB (NF-κB) signaling, in the presence of reduced antioxidant defenses and increased generation of reactive oxygen species [30]. As NF-κB is under epigenetic regulation by histone deacetylases [5], such as NAD^+^-dependent sirtuin 1 [31], this suggests sirtuin 1 agonists, such as resveratrol, may be an attractive therapeutic intervention for ‘burden of lifestyle’ diseases [5]. Notably, it has already been demonstrated to be effective at reducing VC and atherosclerosis in mice [32].

### 2.3. SCAPs

SCAPs have been termed “the Achilles’ heel of senescent cells”. These are anti-apoptotic pathways which exert their actions to protect themselves from their own SASP (i.e., recruiting immune cells to initiate apoptosis) [17]. These include p21, phosphoinositide 3-kinase (PI3K)δ and B-cell lymphoma-extra large (BCL-xL), which were among the first SCAPs to be identified and targeted with the senolytic compounds Quercetin and Dasatinib (often used in combination) to selectively induce apoptosis in senescent cells [33]. Since the original observation, an additional SCAP, based around the activity of heat shock protein 90, has been identified [34]. 

### 2.4. Biomarkers of Cellular Senescence 

Finding universal biomarker(s) for senescence is not straightforward [35]. Differing exposomes and differing life course stages may require different markers to allow for the effects of antagonistic pleiotropy and lifestyle. Burden of lifestyle diseases may also require a marker(s) that can distinguish features of ageing *per se* from features specific to morbidity [5]. Inter-individual differences in age-related physical capability also are significant and may not accurately reflect the burden of cellular senescence, due to number and location of senescent cells differentially affecting function dependent on the organ and tissue in which they reside [36]. 

The molecular pathways inducing cells to a senescent state are still not fully defined [37]. Indeed, an accepted description of what actually constitutes senescence at a molecular level remains debated. Furthermore, various pro-inflammatory cells (e.g., a peritoneal-derived macrophage subpopulation) can be induced to show some features more typically associated with senescence, such as beta-gal staining and p16^ink4a^ expression [38]. While this emphasises the need for more specific biomarkers which fit accepted criteria for a BoA [3], the cognate transcript for p16^ink4a^ termed CDKN2A, has proven robust in vivo, when applied to measures of age-related physiological function in healthy tissue. Analysis of renal allografts has shown CDKN2A expression to be a consistently better predictor of biological age and age-related physiological function than telomere length [39,40,41].

### 2.5. Senescence in CKD–MBD

Abnormalities of mineral metabolism in CKD–MBD manifest from progressive loss of renal function and dysregulated bone metabolism. This gives rise to a toxic uremic milieu, which promotes vascular smooth muscle cell (VSMC) proliferation, migration, apoptosis, and senescence, further inducing a premature ageing phenotype in multiple cell types, most notably in arterial and bone tissue [22]. These clinical sequelae are associated with CVD and osteoporosis, respectively [14]. Compelling novel data suggests that uremia accelerates the accumulation of senescent cells in the kidney, contributing to a decline in renal function in CKD, while removal of senescent cells restores renal function in ageing mice [42]. It is also conceivable that the toxic uremic milieu spreads allostatic load across the body and contributes to senescence of cells in other tissues and organs, such as the skeleton and vasculature [14].

Both developmental and oncogene-induced senescent cells are present in the kidney over the life course, with the deleterious chronic class of these cells postulated as being associated with promoting pathology in acute kidney injury, diabetic nephropathy and CKD–MBD [8]. Oncogene-induced senescence is present in a variety of cell types in the setting of CKD, including stem cells, endothelial cells, immune cells, and progenitor cells [43].

In aged mice, osteoclasts, osteoblasts and osteoclast progenitor cells display increased expression of *p16^INK4a^* in comparison to young mice [23]. Moreover, targeting of senescent cells in aged mice via genetic ablation, SASP inhibitors or senolytic compounds, independently improved bone mass and bone microarchitecture outcomes in all instances [44]. Although this suggests senescent cells have a causal role in the pathogenesis of age-related bone loss, this has not yet been studied in the context of CKD–MBD. The strong independent association between low bone density and increased VC [45] implies an inverse relationship between bone mineral density and early vascular ageing (EVA) associated with arterial stiffening and cardiovascular mortality [46]. 

## 3. Calcification 

### 3.1. Introduction to Calcification 

Calcification as part of EVA is not a recent observation, being first described in the setting of renal disease by Virchow in 1855, while VC *per se* has been described in humans living 5000 years ago [47]. However, in similarity with senescence, much is still to be learned regarding its aetiology. Hydroxyapatite crystal deposition can present in the tunica media, intima or cardiac valves, a complication of CKD–MBD [48] that is strongly associated with high cardiovascular mortality in this patient population [49]. Furthermore, between 15–20% of calcified vessels and valves are ossified [50]. Nonetheless, a direct and causal relationship between VC and all-cause/cardiovascular mortality is not yet proven, as the possibility remains that VC acts as a confounding factor [49]. Traditional Framingham risk factors (e.g., smoking, diabetes, etc.) are insufficient at accounting for the high incidence of cardiovascular mortality in CKD–MBD, so non-traditional risk factors—such as abnormal mineral metabolism, oxidative stress, and inflammation—have also been implicated [18]. Differences in the pathogenic mechanisms underlying CKD–MBD, anatomical/histological locations and clinical outcomes suggests intimal and medial calcification are “distinct entities”; a concept now widely accepted [48]. 

As previously stated, CKD is characterised by a persistent low-grade inflammation, with factors such as dialysis membranes, oxidative stress, microbial factors, uremic toxins, and cellular senescence contributing to the increased inflammatory microenvironment [51]. Inflammation is also a common feature of its clinical sequelae; a driving force behind the promotion of EVA. IL-6 and C-reactive protein are independent predictors of cardiovascular mortality in CKD patients [52,53]. Furthermore, it has been shown that elevated aortic microinflammation is present both in the absence of calcium and phosphate, and before hydroxyapatite deposition in the media layer, suggesting a role in the initial stages of VC [54]. Additional inflammatory components involved in the pathogenesis of tunica media and intima calcification processes are outlined throughout this section. 

### 3.2. Tunica Media Calcification

Tunica media calcification, also known as Mönckeberg calcification, is associated with arterial stiffening (arteriosclerosis), sudden cardiac death and reduced perfusion of cardiac tissue. Under the umbrella term of VC, this entity is most commonly observed in CKD, and constitutes a clinical hallmark of the disease. Similar to bone formation, media calcification can be partitioned into three phases: initiation, nucleation and crystal growth [55], as both VSMCs and osteoblasts differentiate from mesenchymal progenitor cells [47]. 

The tunica media layer is primarily composed of VSMCs, elastin and collagen [56]. Elastin degradation is required for the creation of nucleation sites for matrix vesicles (present in the initiation phase) and the promotion of osteogenic differentiation of VMSCs, through an upregulation of tumour necrosis factor (TNF) [57]. This process is driven by upregulation of matrix metalloproteinase (MMP) expression, most notably MMP-2 and MMP-9 [58], which have been described as part of the SASP [59]. TNF also promotes an upregulation of Runt-related transcription factor 2 (RUNX2), an important regulator of the trans-differentiation of VSMCs, further increasing expression of osteogenic genes, such as osteocalcin and alkaline phosphatase. This process is further implemented when phosphate enters VSMCs through sodium-dependent phosphate transporter 1 (PiT-1), facilitating the detrimental formation of hydroxyapatite crystals in the extracellular matrix [57]. Pit1 blockade and RUNX2 KO mice have been shown to inhibit mineralisation and osteoblast differentiation, respectively, exemplifying their contribution to VC deposition [60,61]. Circulating vascular cells, mesenchymal progenitor cells (including monocytes) and pericytes may also be additional sources of cells subjected to osteoblast-like trans-differentiation [62]. VSMC apoptosis also promotes VC [63], preceding the deposition of hydroxyapatite crystals [64].

Physiologically, there is a balance of mineral metabolism inducers (e.g., phosphate, parathyroid hormone, fibroblast growth factor 23 etc.) and inhibitors (e.g., matrix Gla protein (MGP), fetuin-A, Klotho etc.) [57]. In the uremic environment the balance is tipped towards a pro-calcific microenvironment, as levels of inducers are increased and inhibitors reduced, the consequence of uremic inflammation and oxidative stress, which suppresses expression of Nrf2-Keap1 [65]. Indeed, animal studies suggest that the Nrf2-Keap1 signaling pathway plays an important role in the VC process. Yao et al. [66] have reported that activation of the Nrf2 signaling pathway may prevent hyperphosphatemia-induced VC by inducing autophagy in VSMCs. Moreover, in an in vivo mouse model, activation of Nrf2 by dimethyl fumarate attenuated Vitamin D3-induced VC [67] and resveratrol was shown to ameliorate VC by regulation of Nrf2 and Sirtuin 1 [68]. Finally, hydrogen sulfide (H_2_S), an endogenous signaling molecule with antioxidant properties, attenuated calcification of VSMCs via activation of Nrf2/Keap1 [69]. The pathogenic effects of only a minority of inducers/inhibitors have been demonstrated in vivo in rodent animal models. For example, Klotho-deficient mice show extensive soft tissue calcification accompanying other premature ageing abnormalities [70]. However, the relative influences of the majority of uremic toxins may still be unknown [71]. In this context, it is of interest that Klotho induces Nrf2-mediated anti-oxidant defenses in human aortic smooth muscle cells [72]. 

### 3.3. Tunica Intima Calcification

In contrast to media calcification, cardiovascular outcomes related to intimal calcification are associated with increased risk of plaque rupture, thereby leading to acute occlusion of vessels [48]. This occurs in medium to large sized arteries on the proximal end of the vascular tree, as opposed to resistance-sized arteries. Key mechanistic differences drive intimal calcification, when compared to medial calcification. For example, atherosclerotic plaques are a pre-requisite, as the deposition of calcium crystals form in the advanced stages of the atherosclerosis [49]. Atherosclerosis, an inflammatory driven disease resulting from endothelial dysfunction [48] begins with accumulation of low-density lipoprotein (LDL) in the artery wall, progressing to form a plaque. The plaque evolves from a small lesion into a clinically dangerous and structurally unstable form [73]. As Nrf2 deficiency promotes plaque instability in hypercholesterolemic mice [74] it is unsurprising that inflammation plays a more prominent role in initiating this process. Macrophages, key players in the atherosclerosis process, release matrix vesicles supplying a nidus for mineralisation within the plaque, and downregulate matrix Gla-protein (MGP) expression [75]. Macrophages also contribute to pro-calcifying pathways by expressing osteogenic proteins RUNX2 and bone morphogenetic protein 2 (BMP2) [76]. Moreover, macrophages are capable of differentiating into osteoclast-like cells, mirroring their mineral resorption function in VSMCs [77]. VSMCs from the neighbouring medial layer also infiltrate the plaque in response to the atherosclerotic lesion [49], a source of the osteogenic mediators, as well as circulating and resident stem cells [78]. 

However, any clinical significance remains debated. As it has been suggested that calcification of plaques prone to rupture may stabilise the plaque [79], VC has been regarded as beneficial in non-renal patients [80]. The clinical implications of this phenomenon in CKD are undetermined, as plaque stability after microcalcification and macrocalcification may differ. Macrocalcification (or spotty calcification), promoted by pro-inflammatory M1 macrophages during atherosclerotic plaque progression, was shown to associate with plaque rupture [81]. Although the exact mechanism(s) of the M1-mediated process remains to be fully elucidated, pro-inflammatory cytokines TNF and oncostatin M may promote osteogenesis by upregulation of RUNX2 and alkaline phosphatase. A healing process is initiated in response, characterised by a phenotypical switch of M1 to M2 macrophages. The anti-inflammatory actions of M2 macrophages promote the resolution of inflammation and maturation of VSMCs—hallmarks of macrocalcification, which is associated with plaque stability [82]. Taken together, the clinical implications of plaque stability are largely dependent on the differential effects of macrophage polarity. 

## 4. Does Senescence Drive Vascular Calcification?

The bulk of current knowledge, relating to senescence as a cause of EVA, has been derived from studies of intima calcification. Hence the association between senescence and VC remains unproven and any findings must be translated to medial calcification with caution. Senescence has been observed in VSMCs at the site of atherosclerotic plaque, promoting plaque formation and vulnerability [83]. Senescence in vascular endothelial cells, endothelial progenitor cells, and VSMCs has been implicated in the development and progression of atherosclerosis [84,85,86]. During the early stages of plaque initiation in a mouse model of atherosclerosis (Ldlr KO) senescent macrophage foam cell accumulation was observed in the sub-endothelial space within the artery wall. It has been suggested that senescent endothelial cells drive atherosclerosis through increased infiltration of oxidized LDLs across a leaky membrane, or via upregulation of MMPs from the SASP, thereby transitioning stable atheromas into unstable atheromas [36]. Evidence regarding the involvement of senescent cells in calcification (i.e., the latter stages of plaque progression) has been reported by Roos et al., [87] through targeting senescent cells with a genetic elimination strategy and senolytic intermittent therapy (Dasatinib + Quercetin) in aged and hypercholesterolemic mice. Senolytic treatment reduced senescence markers in the medial layer of the aorta and reduced aortic calcification in both aged and hypercholesterolemic mice when compared to vehicle-treated mice. This was also associated with reduced protein levels of osterix, an established marker of osteogenesis. 

Burton et al. [88] have proposed a strong link between VSMC senescence and the progression of atherosclerosis (plus the development of VC) based on microarray transcriptome analysis. They demonstrated differential regulation of genes with inflammatory (*IL-8, IL-1β, ICAM1, TNFAP3, ESM1*, and *CCL2*), tissue remodeling (*VEGF, VEGFβ, ADM*, and *MMP-14*) and calcification (*MGP, BMP2, SPP1, OPG*, and *DCN*) functions in senescent VSMCs versus non-senescent proliferating VSMCs. Alique et al. [89] have determined in vitro that plasma microvesicles (MVs) from elderly subjects and from senescent endothelial cells promote VC in human aortic smooth muscle cells. Senescent human umbilical vein endothelial cell (HUVEC)-derived MVs and MVs from elderly subjects expressed more pro-calcification proteins (e.g., annexin A2, annexin A6 and BMP2) and increased calcium content, when compared to young cells. It has been suggested that differences between contractile and synthetic VSMCs may reflect their origin, rather than trans-differentiation from a common source [86]. Based on this hypothesis, findings relating to senescent VSMCs at the site of plaque rupture indicate that senescence in circulating progenitor cells contribute to the pathology. However, calcification was not examined in this study [78].

Studies specific to CKD have also been completed, albeit to a lesser degree. We have reported previously a positive correlation between CDKN2A/p16^INK4a^ and medial calcification in uremic epigastric arteries. In addition, CDKN2A/p16^INK4a^ and SA-β-Gal expression increased with the degree of medial calcification, suggesting this may contribute to the pathogenic mechanism(s) underlying VC [9]. Despite the evidence presented, many questions remain to be answered before a causal relationship between senescence and EVA can be established. For example, is the senescence and VC relationship bidirectional, as determined experimentally with inflammation and VC? Inflammation is a well-defined risk factor underlying the pathogenesis of VC in CKD, however it has been demonstrated that micro-calcification may drive further vasculature inflammation in carotid arteries, highlighting the importance of investigating the possibility of senescence as a consequence and inducer of VC [90]. 

## 5. Targeting Vascular Calcification Therapeutically

At present, clinical therapeutic interventions to reverse VC remain in development [49]. While non-calcium phosphate binders can slow VC progression, they are not curative [91]. Targeting some of the aforementioned inducers and inhibitors, which have been found to be elevated or reduced in CKD patients, is an ongoing approach to intervene in disease progression, as seen with vitamin D supplementation. Analogous to statin therapy and renal transplantation, this approach has not yielded convincing data on VC regression. Based on the growing body of exciting pre-clinical data, targeting removal of senescent cells in this context remains a possible avenue for achieving significant intervention. Senolytic compounds are thus an attractive therapy with the potential to address multiple ‘burden of lifestyle’ diseases where senescence may be an underpinning component. In support of such a hypothesis, data from Xu et al. [16] has demonstrated prolongation of the lifespan of naturally aged mice and senescent cell transplanted young mice by 36%, using a senolytic combination of Dasatinib and Quercetin. The same ‘cocktail’ of senolytic drugs attenuated physical dysfunction and reduced VC in the aorta of aged and hypercholesterolemic mice when administered long-term [87]. Despite the promising preclinical findings, a number of unknowns remain, such as undefined side effects and the optimal time to initiate treatment in the patient. It also remains to be seen if senolytic cocktails are effective in reducing VC in the uremic toxic milieu with high phosphate and reduced Klotho expression, as to the best of our knowledge, no such studies have been conducted. 

In the aforementioned mouse model studies, the following drug concentrations were used: 5 mg/kg Dasatinib and 10–50 mg/kg Quercetin, or an isovolumic vehicle, via oral gavage [16,87]. Quercetin, a flavonoid with anti-inflammatory actions, can be consumed through diet or supplementation, however the synergistic actions of Dasatinib and Quercetin combined were shown to be superior in removing senescent cells than either compound alone [33]. In a phase II clinical trial for senescence in CKD (currently in its recruitment phase), the intervention arm consists of a daily dosage of 100 mg Dasatinib and 1000 mg Quercetin [92], equating to 1.43 mg/kg and 14.3 mg/kg for a 70 kg human, respectively. Future clinical trials will determine the tolerability and efficacy of concentrations physiologically achievable in humans. 

An alternative to the ablation of senescent cells is to target the SASP. Hegner et al. [93] have ameliorated uremia-induced calcification in vascular progenitor cells in vitro by pharmacologically blocking pro-inflammatory cytokines. Although markers of senescence were not studied, the known SASP components TNF and IL-1β were among the cytokines altered. Finally, as activation of the Nrf2 signaling pathway has inhibited VC in several experimental animal models [66,67], the effects of Nrf2 agonists, such as bardoxolone methyl and dimethyl fumarate, on arterial calcification and stiffness should be tested. As Nrf2 may control different mechanisms involved in the physiopathology of bone disease [94], Nrf2 activation may also be a novel therapeutic strategy for uremic bone disease.

## 6. Conclusions

While the involvement of senescence in VC, bone disease, and other ‘burden of life style’ diseases is beginning to be appreciated from proof-of-concept studies, true causation can only be proven by randomised double-blind controlled studies. Further research into understanding how risk factors and inducers drive mechanisms, which result in VC is needed before therapeutic interventions become a viable option. Prior to this, critical issues of efficacy and safety should be assessed for successful development and implementation. However, since therapeutics targeting senescent cells and the Nrf2 signaling pathway are beginning to mature, studies need to be conducted to unravel if these treatment strategies may offer a novel possibility to interfere with the progeric phenotype that results as a consequence of CKD.

## Figures and Tables

**Figure 1 toxins-11-00082-f001:**
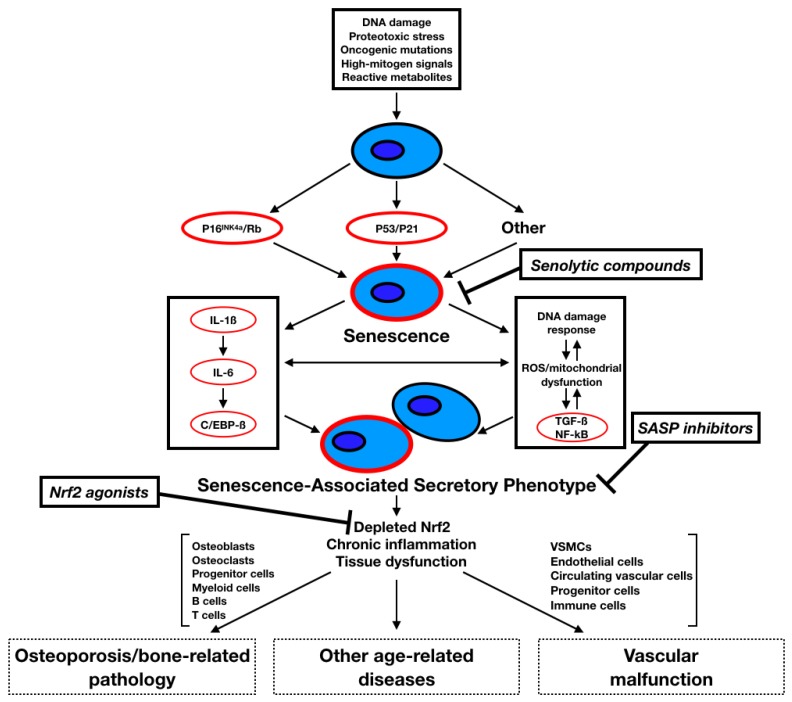
A schematic view of the pathways, molecular mechanisms and cell types inducing cellular senescence, thus leading to chronic age-related diseases. C/EBP-β, CCAAT/enhancer binding protein-β; IL-1β, interleukin 1β; IL-6, interleukin 6; NF-κB, nuclear factor κB; ROS, reactive oxygen species; TGF-β, transforming growth factor β; nuclear factor-erythroid-2-related factor 2, Nrf2. Adapted from Tchkonia et al. 2013 [19].

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
