# Peer review of "Senescent Cells in Early Vascular Ageing and Bone Disease of Chronic Kidney Disease—A Novel Target for Treatment"

_toxins, 2019, doi:10.3390/toxins11020082_

Reviewer 1 Report

~Line 299: Authors should include a discussion on the content of the senolytic “cocktail.” What was the concentration and route of administration of the cocktail in these mice? Are these concentrations physiologically relevant/achievable in humans? Discuss why this treatment would be more beneficial to reduce prevelance of senescent cells in CKD over trying to eat foods high in these phytochemicals. I.e. are these concentrations not achievable through the diet?

Authors need to include a discussion on how effective reducing senescent cells would be on reducing vascular calcification in context of rising serum P and deficient klotho (very well researched area in the context of vascular calcification). Neither of these two factors would be absent, even given a senolytic cocktail. Need to discuss this complexity and discuss any work that has been done to mitigate serum P while also addressing premature again cells.

Author Response

We thank the reviewers and editor for their helpful comments which we have addressed in the revised manuscript. This document outlines our response and changes made accordingly. We believe the manuscript has been improved by the amendments, and look forward to your answer regarding acceptance for publication.

Reviewer 1

~Line 299: Authors should include a discussion on the content of the senolytic “cocktail.” What was the concentration and route of administration of the cocktail in these mice? Are these concentrations physiologically relevant/achievable in humans? Discuss why this treatment would be more beneficial to reduce prevalence of senescent cells in CKD over trying to eat foods high in these phytochemicals. I.e. are these concentrations not achievable through the diet?

Our comment: All above questions have been addressed in text (line 334 onwards).

Authors need to include a discussion on how effective reducing senescent cells would be on reducing vascular calcification in context of rising serum P and deficient klotho (very well researched area in the context of vascular calcification). Neither of these two factors would be absent, even given a senolytic cocktail. Need to discuss this complexity and discuss any work that has been done to mitigate serum P while also addressing premature again cells.

Our comment: To the best of our knowledge there are no studies that have tested if senolytic cocktails are effective in reducing VC in the toxic uremic milieu with high phosphate and reduced klotho expression. In the revised version of the manuscript this caveat is mentioned (line 331 onwards).

Reviewer 2 Report

This is a well written review on an important topic in CKD. It reads well and is organized thoughtfully

Author Response

We thank the reviewers and editor for their helpful comments which we have addressed in the revised manuscript. This document outlines our response and changes made accordingly. We believe the manuscript has been improved by the amendments, and look forward to your answer regarding acceptance for publication.

Reviewer 3 Report

Study Summary 

This review highlighted the role of cellular senescence and its presumed role in vascular calcification in the setting of chronic kidney disease. The Review is very well organized and majority of references are within past 5 years and the sections are clear and detailed. 

Major

This review could benefit from adding a section to discuss the role of inflammation as inflammation plays a major role in vascular calcification, progression of CKD and in accelerating the aging process. Similarly, the title mentions identification of a “novel target” but the review would benefit from a more explicit thesis highlighting the common molecular mechanisms/pathways which link early vascular ageing with CKD-MBD and which constitute the novel target, assuming that this target is a common pathway which links the two. This could occur in both the summary Figure and/or the introduction.

Minor

1) It would be good to note the capability of macrophages to produce key osteogenic proteins Runx2 and BMP-2 for roles other than release of matrix vesicles. See the following references:

a) Reduced plaque calcification and volume in LDL-receptor knockout mice with macrophage-specific loss of TRPC3. Atherosclerosis 2018 270, 199-204. 

b) Evidence for constitutive bone morphogenetic protein-2 secretion by M1 macrophages: constitutive auto/paracrine osteogenic signaling by BMP-2 in M1 macrophages. Biochemical and Biophysical Research Communications 2017491, 154-158. 

2) The authors mention how calcification relates to plaque stability which stands true in settings of macro-calcification , however micro-calcification is known to cause plaque instability. It would be good to discuss few sentences to compare and contrast  micro-and macro-calcification in this regard.

3) There is a typo for BMP2 in line 272.

Author Response

We thank the reviewers and editor for their helpful comments which we have addressed in the revised manuscript. This document outlines our response and changes made accordingly. We believe the manuscript has been improved by the amendments, and look forward to your answer regarding acceptance for publication.

Reviewer 3

Major

This review could benefit from adding a section to discuss the role of inflammation as inflammation plays a major role in vascular calcification, progression of CKD and in accelerating the aging process. Similarly, the title mentions identification of a “novel target” but the review would benefit from a more explicit thesis highlighting the common molecular mechanisms/pathways which link early vascular ageing with CKD-MBD and which constitute the novel target, assuming that this target is a common pathway which links the two. This could occur in both the summary Figure and/or the introduction.

Our comment: As the review focuses on early vascular ageing, additional inflammatory mechanisms promoting vascular calcification have been added. Due to the format of the section (i.e. tunica media and tunica intima calcification subsections), additional text has been added to the relevant subsection, as opposed to an “Inflammation” subsection per se. Therefore, please see ‘Section 3: Calcification’, where stated in tracked changes.

The diagram has been amended to highlight the common pathways, with the addition of three therapeutic interventions and their novel targets.

Minor

1) It would be good to note the capability of macrophages to produce key osteogenic proteins Runx2 and BMP-2 for roles other than release of matrix vesicles. See the following references:

a) Reduced plaque calcification and volume in LDL-receptor knockout mice with macrophage-specific loss of TRPC3. Atherosclerosis 2018 270, 199-204. 

b) Evidence for constitutive bone morphogenetic protein-2 secretion by M1 macrophages: constitutive auto/paracrine osteogenic signaling by BMP-2 in M1 macrophages. Biochemical and Biophysical Research Communications 2017491, 154-158. 

Our comment: This has been addressed in text. Please see from line 252.

2) The authors mention how calcification relates to plaque stability which stands true in settings of macro-calcification, however micro-calcification is known to cause plaque instability. It would be good to discuss few sentences to compare and contrast micro-and macro-calcification in this regard.

Our comment: This has been addressed in text. Please see from line 260.

3) There is a typo for BMP2 in line 272

Our comment: This has been addressed in text. Please see line 294.

Round  2

Reviewer 1 Report

Authors have adequately addressed my concerns